# An investigation into the relationship between multimedia lecture design and learners' engagement behaviours using web log analysis

Cailbhe Doherty⬤[1,2]*

1 School of Public Health, Physiotherapy and Sports Science, University College Dublin, Dublin, Ireland,
2 Insight Centre for Data Analytics, University College Dublin, Dublin, Ireland

* cailbhe.doherty@ucd.ie

## Abstract

The purpose of this transaction log analysis was to evaluate university students' engagement behaviours with a catalogue of multimedia lectures. These lectures incorporated selected instructional design principles from the cognitive theory of multimedia learning (CTML). Specifically, thirty-two multimedia lectures which differentially employed the signalling, segmenting and embodiment principles from the CTML were delivered to a cohort of 92 students throughout an academic trimester. Engagement with each multimedia lecture was measured in three domains: affective engagement was measured using a Likert-style survey that accompanied each multimedia lecture; behavioural engagement was measured using the web logs provided by YouTube Studio analytics (average watch time); cognitive engagement was measured using students' average score on a quiz that accompanied each multimedia lecture. Separate multiple linear regression analyses for measures of affective, behavioural and cognitive engagement revealed that multimedia lectures that 'stacked' the instructional design principles of embodiment (whereby the lecture was interspersed with clips of an enthusiastic onscreen instructor), segmenting (where lectures were divided into shorter, user-paced segments) and signalling (where onscreen labels highlighted important material) increased measures of engagement, including overall watch time, number of survey submission and number of quiz attempts ($P < 0.05$). There was no association between any of the tested principles and students' quiz scores or their responses on the Likert-style survey. This study adds to the available literature demonstrating the effectiveness of the signalling, segmenting and embodiment principles for increasing learner engagement with multimedia lectures.

## Introduction

In recent years, the multimedia video lecture has gained prominence as a teaching format in university education [1], a trend that was accelerated during the COVID-19 pandemic [2]. The widespread adoption of multimedia video lectures and other "asynchronous" learning material

**Data Availability Statement:** The data for this investigation have been made shareable and can be accessed on OSF.io using the following link: https://osf.io/9aerk/.

**Funding:** The author received no specific funding for this work.

**Competing interests:** The author has declared that no competing interests exist.

has been extensively documented in prior research [3–8]. While some educators have expressed concern that pre-recorded multimedia lectures are not as effective as face-to-face instruction [9, 10], a large body of empirical research now advocates their effectiveness as educational tools [3–8]. Importantly, the effectiveness of multimedia lectures is predicated on good course content and pedagogy, when multimedia instruction is grounded in a sound theoretical framework that is supported by empirical evidence [11, 12].

## The Cognitive Theory of Multimedia Learning

One such framework is the Cognitive Theory of Multimedia Learning (CTML), which has evolved within a body of research papers and books produced by Mayer and colleagues over the past 30 years [13]. Although its name has changed over time, the theory is based on the premise that humans possess two different channels for processing material, one for visually based representations and one for verbally based representations [14]. Each channel has limited processing capacity, so people learn better when the two channels are used together to help them make sense of presented material [11, 14]. Stated more simply, the theory posits that people learn better from words and graphics than from words alone.

According to the CTML and the cognitive load theory [15] on which it is based, learners can engage in three kinds of cognitive processing during learning, each of which draws on the their available cognitive capacity: 1) extraneous processing, 2) essential processing, and 3) generative processing. Briefly, extraneous processing refers to cognitive processing that does not serve the instructional goal [14]. For example, if a medical student was presented with a model of the human muscular system in which the labels for corresponding anatomical structures were unclear, they would have to engage in extraneous cognitive processing to discern which labels were linked with the different parts of the illustration. Essential processing refers to cognitive processing aimed at mentally representing the learning material in working memory. Essential processing is affected by the complexity of the material, the quantity of information being presented or its overall duration. Generative processing refers to cognitive processing aimed at making sense of the presented material and is linked with the learner's motivation to learn [14]. For example, when the material is presented by an enthusiastic instructor, the learner may exert more effort to understand what the instructor is presenting.

**Using the Cognitive Theory of Multimedia Learning in instructional design.** It follows that there are three kinds of instructional design objectives that correspond to the three kinds of demands on cognitive capacity: 1) reduce extraneous cognitive processing, 2) manage essential cognitive processing, and 3) foster generative cognitive processing [14].

The CTML proposes fifteen principles that can be used to reduce extraneous cognitive processing, manage essential cognitive processing, and foster generative cognitive processing [13]. For example, on screen labels can be employed to highlight the most important material for the learner, thereby reducing extraneous processing [16–18]. This is an example of the "signalling principle" [19]. To manage essential processing, videos can be split into shorter chunks for the learner [20]. This is an example of the "segmenting principle" [21]. To foster generative processing, an instructor can be presented intermittently throughout a lecture as a "talking head", encouraging learners to exert more effort to understand what the instructor is presenting [20]. This is an example of the "embodiment principle" [22].

By reducing extraneous cognitive processing, managing essential cognitive processing and fostering generative cognitive processing in the learner, the opportunity for knowledge retention and transfer is optimised [14].

**Gaps in the research.** Although the CTML is a useful framework for guiding instructional design, the available research has failed to clarify several important issues. These issues undermine educators' capacity to apply CTML research in practice and pedagogy.

First, principles from the CTML tend to be evaluated individually, however this does not reflect "real world" scenarios where learning materials will often incorporate several principles (like the signalling, segmenting and embodiment principles) together in a single piece of multimedia [23]. Extant research also rarely evaluates this kind of content in real-world contexts, where learning material is part of a formal degree program [20].

Second, the CTML literature tends to categorically define each principle in an "all or nothing" way. This creates a false dichotomy, distorting the association between a principle and the experimental outcome. For example, the embodiment principle states that people learn more deeply from multimedia presentations when an onscreen instructor is present who engages in human-like gestures as they explain a topic or theory [22]. Numerous studies have shown that students perform better on a transfer test after learning from a high-embodied instructor than a low-embodied instructor or with no onscreen instructor [24, 25]. However, the boundaries of the positive relationship between embodiment and knowledge retention are unclear; the available literature cannot elucidate the relationship between learning from a high-embodied instructor who is present for 10-seconds at the start of a 10-minute multimedia lecture compared with if they are present throughout.

A final limitation with the available research is that the quality and effectiveness of instructional multimedia is generally evaluated using subjective learner interviews and surveys [26–29]. In adopting these obtrusive approaches to directly elicit data from research participants, important insights are likely being missed regarding actual user behaviour. With every page view, followed link or watched video, learners are leaving a measurable trace of their engagement with web-based multimedia, and these behaviours may be altered under direct observation [30].

## Transaction log analysis

Advancing on traditional measures like interviews and surveys, transaction log analysis is a broad methodological approach used to evaluate the electronic records of interactions that have occurred between a system and users of that system [31]. Alternately referred to as weblog analysis, search log analysis and digital trace data analysis [32], transaction log analysis has been in use since at least 1967 [33] and has been published in peer reviewed research since 1975 [34], this methodology has undergone a resurgence in recent times having been made more accessible by the widespread availability of user-facing analytics APIs by video sharing and social media platforms like YouTube and Facebook.

For transaction log analysis, behaviour is the essential construct and is defined as any observable activity of a person, team, organization, or system; session duration or number of clicks are typical variables that researchers are interested in [35]. From the perspective of measuring learners' behaviour, the advantage of transaction log analysis over interviews, surveys or think-aloud exercises is that it is an unobtrusive method of data collection as it does not involve directly interfacing with learners [31].

Transaction logs are widely captured in university settings: e-resource usage, computer logins, attendance and activity in virtual learning environments have been evaluated to discern how students learn [36]. Despite this, universities have been slow to fully utilise the potential of transaction log analysis to shine a light on information-interaction behaviours on content delivery platforms such as YouTube. Despite all the benefits of the Web for eLearning [37, 38], social media-based education [39], virtual reality classrooms [40] and as a channel for

academic institutions to supply their students with content in a socially distant society [41], there is a dearth of research that leverages transaction log analysis methods to evaluate user behaviour during multimedia instruction, particularly when evaluated under the paradigm of the CTML.

### Aims of this research

In summary, there is currently a lack of research evaluating the effect of combining multiple principles from the CTML together to reduce extraneous, manage essential or foster generative cognitive processing among learners as they consume multimedia content; that research also tends to evaluate each of instructional principles in a categorical way, creating a false dichotomy between the tested principle and the objectives of instructional design; the available research usually relies on obtrusive data acquisition methodologies which are distant surrogates of actual user behaviour.

Therefore, the aim of this study was to leverage transaction log analysis methodologies to evaluate how several of the instructional design principles from the CTML affect actual user behaviour in an authentic educational setting. To do so, a catalogue of asynchronous multimedia lectures was developed. This catalogue of video lectures differentially employed, to greater and lesser extents, instructional design principles from the CTML including the signalling, segmenting, and embodiment principles. Transaction log analysis was then combined with survey data and quizzes to evaluate user behaviour and learning during each multimedia lecture.

Our hypotheses were as follows:

1. There is a direct relationship between the extent to which an instructional design principle is incorporated in a multimedia lecture (e.g., more labels) and actual user behaviour (e.g., watch time).

2. Multiple principles can be 'stacked' together (e.g., labels can be combined with a high embodied instructor) to further alter user behaviour during multimedia instruction.

3. User behaviour during multimedia instruction is associated with knowledge retention.

## Materials and methods

This section outlines the research methods for this work. Firstly, the experimental design is specified. Next, the pedagogical aspects, including how the catalogue of multimedia lectures were developed to incorporate selected principles from the CTML are described. Then the technical aspects, including how the lectures were shared with students and their engagement behaviours with them monitored, are discussed. The demographics of the learners' evaluated in the experiment and the methods of data collection are subsequently reported. Finally, the data analyses that were used to answer the research questions are described.

This study was approved by the ethics committee of the institution at which the primary author was based (*ref*: LS-20-47). The need for consent was waived by the ethics committee because no identifiable data about individual participants was captured (only aggregate data).

### Experimental design

We employed the methodology of transaction log analysis [31] to evaluate learners' engagement behaviours during multimedia instruction. Transaction log analysis is a broad categorisation of methods including Web log analysis (i.e., analysis of Web system logs), blog analysis, and search log analysis (analysis of search engine logs). A transaction log is an electronic record of interactions that have occurred between a system and users of that system [31]. As

such, transaction log analysis enables macro-analysis of aggregate user data. In this study, we used the web logs provided by YouTube Studio analytics [42] to evaluate learners' engagement behaviours during multimedia instruction, including audience retention rates for each multimedia lecture. These are described in further detail in the 'technical aspects' section.

## Pedagogical aspects

The development of multimedia lectures for each topic and lecture followed a series of steps. First, a lecture script and accompanying storyboard were prepared for each topic. A teleprompter system was setup, and an instructor was recorded as they read the script aloud. Audio input was recorded via a microphone (Yeti, Blue Designs Inc, CA, USA) connected to a laptop and video input was recorded (1080p at 25 frames per second) using a Canon 250d DSLR camera (Canon Inc, Tokyo, Japan). Next, accompanying videos and slides were developed based on the original storyboard. Both the audio and video media were then imported into a commercially available video editing software (Final Cut Pro; Apple Inc, CA, USA) and time-aligned. The audio and video recordings were trimmed to remove mistakes (e.g. mispronunciations) and large gaps. Standardised logos, colours, formatting and sequencing were applied over the video media to establish a recognisable profile for the multimedia lectures [11].

Next, selected multimedia instructional principles [13] were differentially applied to the multimedia lectures. Specifically, for the signalling principle, on screen labels were used in varying amounts within each multimedia lecture to highlight important material for the learner [19]. For the embodiment principle, a video of the instructor enthusiastically explaining relevant topics or theories was presented intermittently throughout each lecture, in varying amounts [22]. For the segmenting principle, lectures were split into 'chunks' of shorter duration [21]. Lectures with higher levels of signalling or embodiment had more labels or clips of the instructor interspersed throughout, respectively. For example, the range of time during which labels were present on screen in the catalogue of multimedia lectures was between 13–85% of their overall duration. The range of time during which a high embodied instructor were present on screen in the catalogue of multimedia lectures was between 0–60% of their overall duration. Lectures that were split into shorter chunks were deemed to represent 'higher' levels of segmentation. The shortest multimedia lecture was 57 seconds in duration and the longest was 13 minutes and 58 seconds in duration.

In summary, this process resulted in the creation of a catalogue of thirty-two multimedia lectures covering twelve topics which incorporated each of the selected multimedia instructional principles on a continuous scale. Examples of how these principles were followed are displayed in Fig 1.

## Technical aspects

YouTube (YouTube Inc, CA, USA) was chosen as the platform to house the catalogue of multimedia lectures and share them with learners throughout the academic semester. The advantages of YouTube as a video sharing platform is that it is free, content sharing can be restricted to specific groups and because it provides useful web log analytics about user engagement behaviours [42]. Specifically, the meta-usage data of interest included the number of views accrued for each video and audience retention, which indicates when viewers start or stop watching a multimedia lecture and for how long they watch it.

After development, the catalogue of multimedia lectures were uploaded to YouTube with 'private' visibility, whereby the primary author had sole access rights. During the week of term that the lecture was scheduled to be delivered to students, the visibility for that lecture was

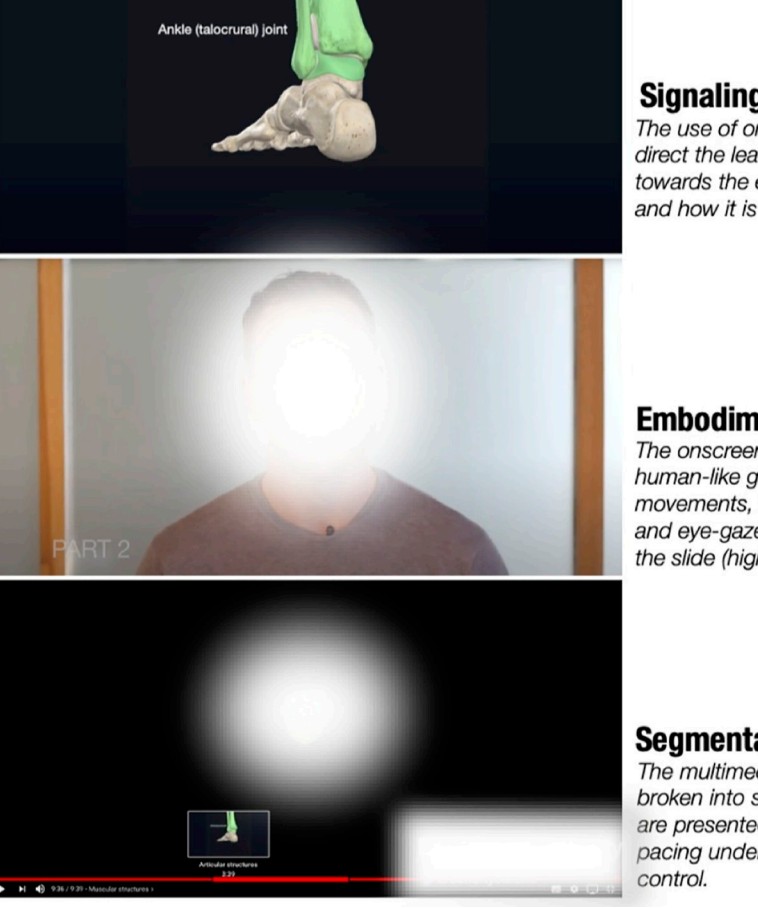

**Fig 1. An example of the signalling, embodiment and segmentation principles for one of the multimedia lectures included in the analysis that related to the functional anatomy of the ankle complex.**

altered to 'unlisted' and students were provided with a link to view the multimedia lecture for a 1-week period as part of a learning 'unit' via the university's virtual learning environment (VLE), Brightspace (D2L Europe Ltd). Each learning unit consisted of the multimedia lecture for that topic, a topic-specific multiple choice questionnaire (MCQ) and a single-item Likert survey where students were asked to rate their agreement with the statement "This lecture enhanced my learning/understanding of course content" on a scale from 1 ("strongly disagree") to 5 ("strongly agree"). The quizzes that accompanied each multimedia lecture were comprised of between 3–5 questions covering the material that was presented in that lecture. None of students' behaviours, their responses on the Likert survey or their scores on the MCQs contributed towards their overall grade in the module, however students were informed that the MCQ questions would be similar to those included in their summative end-of-semester MCQ and practical assessment.

The learning unit was set up using the VLE's restriction options such that students could only access and complete the MCQ and survey after they had accessed the associated multimedia lecture. MCQs and surveys were only accessible in the topic week, after which they were made unavailable. A 'live-lecture' was delivered to students by the lecturer at the beginning of

the semester outlining the learning outcomes for the module and the method of delivery of learning material (i.e. through the learning units), and each week following to review that week's topic and to facilitate class discussion.

## Cohort of learners

The study site was the University at which the primary author is based. The module for which instructional multimedia was prepared was entitled 'Functional Anatomy & Kinesiology' (12 topics, 32 lectures), a module delivered to first-year students enrolled in the University's Physiotherapy and Sports Science Programmes. The purpose of this module is to facilitate students in gaining an understanding of the structure and function of normal human anatomy while providing a basis for their understanding and analysis of posture and normal human movement. It encourages students to apply their knowledge of anatomy in a practical setting.

Ninety-two students enrolled in the module (34 males, 58 females; mean age = 20.2yrs [95% CI = 19.6 to 20.8yrs]) in the 2020–2021 Spring semester, gaining access to the catalogue of multimedia lectures sequentially over a period of 12-weeks from 18th January to the 12th of April 2021. The period of observation for each learning unit (i.e. the multimedia lecture, the quiz associated with each lecture and the Likert-style survey) was one week. Once a one-week interval had elapsed, the surveys and quizzes were removed from the VLE and each multimedia lecture was made permanently available to students in non-segmented form for revision purposes; these non-segmented versions of the multimedia lectures were not included in the current analysis. The learning units were only delivered to students who had registered for this module.

## Outcomes and data analysis

Engagement behaviours with the multimedia lectures was assessed in three domains [43]: 1) Affective engagement was assessed via the number of submissions and the median score on the single-item Likert survey that accompanied each multimedia lecture; 2) Behavioural engagement was quantified based audience retention (which indicates when viewers start or stop watching a multimedia lecture and for how long), a metric provided by YouTube studio [42]; 3) Cognitive engagement was assessed via the number of submissions and the average score in the MCQ included in each learning unit [26]. These experimental outcomes are defined in Table 1.

Descriptive statistics were used to summarize multimedia lecture characteristics and aggregate-level engagement behaviours with these lectures. Continuous variables are expressed as means or medians, and standard deviations (sd).

Separate multiple regression analyses were run to predict 1) Affective engagement (the number of Likert-style survey submissions); 2) Behavioural engagement (audience retention); 3) Cognitive engagement (number of quiz submissions and average quiz scores) based on the extent to which the segmenting, signalling and embodiment principles were implemented within the catalogue of multimedia lectures. A cumulative odds ordinal logistic regression with proportional odds was run to determine the effect of the same principles (i.e., segmenting, signalling and embodiment) on the median group score on the Likert style survey that accompanied each multimedia lecture.

The assumptions for multiple regression were assessed and data distributions were visually inspected for each outcome measure. For each regression model, there was linearity as assessed by partial regression plots and a plot of studentized residuals against the predicted values. There was independence of residuals based on a Durbin-Watson statistic. There was homoscedasticity based on visual inspection of a plot of studentized residuals versus unstandardized predicted values. There was no evidence of multicollinearity as assessed by tolerance values greater than 0.1. There were no studentized deleted residuals greater than ±3 standard

**Table 1. The complete list of and definitions of the experimental outcomes, including the multimedia characteristics and measures of engagement.**

| | | |
|---|---|---|
| Multimedia characteristics | Signalling | The amount of time signals, defined as on screen visual cues (such as arrows, or short pieces of text), were present on screen. Expressed as a percentage of overall multimedia lecture duration. |
| | Segmenting | The length of the multimedia lecture (in seconds). |
| | Embodiment | The amount of time the lecturer was presented on screen, enthusiastically demonstrating learning material. Expressed as a percentage of overall multimedia lecture duration. |
| Cognitive engagement | Quiz attempts | The number of attempts completed by learners on the quiz associated with a multimedia lecture. Expressed as a percentage of the number of learners who had access to the quiz (i.e. the total number of learners who had access to the learning units [N = 92]). Each student could only attempt each quiz once. |
| | Quiz scores | The average score achieved by learners on a quizzes that accompanied each multimedia lecture. |
| Behavioural engagement | Audience retention | The average view duration for a multimedia lecture, expressed as a percentage of overall multimedia lecture length. |
| Affective engagement | Survey submissions | The number of submissions completed by students on the Likert survey associated with a multimedia lecture. Expressed as a percentage of the number of students who had access to the survey (i.e. the total number of learners who had access to the learning units [N = 92]) |
| | Median survey score | The median score on a Likert-survey that accompanied each multimedia lecture in which students were asked to respond to the statement "This lecture enhanced my learning/understanding of course content" on a scale from 1 ("strongly disagree") to 5 ("strongly agree"). |

Note: audience retention values can exceed 100% when learners watch a multimedia lecture (or parts of a multimedia lecture) multiple times.

deviations, no leverage values greater than 0.2, and values for Cook's distance above 1. The assumption of normality was met for each model (as assessed by a Q-Q Plot). The alpha level for each multiple regression analysis was set to $P < 0.05$.

## Data access

Data collected during the current experiment can be accessed using the following link to open access repository, OSF.io: https://osf.io/kpgde/?view_only=9cb8326ddb18443f9caa727d5838e88b

## Results

During the observation period, the multimedia lectures were collectively viewed 2002 times (mean = 63 views per multimedia lecture; 51 unique viewers per multimedia lecture). The average view duration was 90% of the overall multimedia lecture length (range 64–112%). There were a total of 1533 quiz attempts (an average of 49% of the cohort of learners) and the average score on the quizzes was 69%. There were 841 survey submissions (an average of 26% of the cohort of learners) and the median score on the Likert-survey (in which students were asked to respond to the statement "This lecture enhanced my learning/understanding of course content" on a scale from 1 ["strongly disagree"] to 5 ["strongly agree"]) was 4.25.

Descriptive statistics of learners' engagement behaviours with the multimedia lectures (including the number of survey submissions, the median survey score, the audience retention, the number of quiz attempts and the average quiz scores for each multimedia lecture), in addition to their characteristics (i.e., the extent to which they incorporated the segmenting, signalling and embodiment principles) are presented in Table 2.

### Association between multimedia lecture design with affective, behavioural and cognitive engagement behaviours

**Affective engagement.** The multiple regression model statistically significantly predicted the number of attempts at each quiz that accompanied each lecture, $F_{(3, 31)} = 5.426$, $p = .005$,

**Table 2. The multimedia lecture characteristics and descriptive statistics for affective, behavioural and cognitive engagement.**

| | Week | Segment | Duration (s) | Signals | Embodiment | Affective | | Behavioural | Cognitive | |
|---|---|---|---|---|---|---|---|---|---|---|
| | | | | | | Survey submissions | Median survey Score | Audience retention | Quiz attempts | Average quiz scores |
| Fundamental principles of Functional Anatomy | 1 | Part 1 | 290 | 62% | 44% | 50% | 4.5 | 83% | 71% | 63% |
| Fundamental principles of Functional Anatomy | 1 | Part 2 | 356 | 31% | 44% | 41% | 4 | 90% | 62% | 74% |
| Postural assessment | 1 | Part 1 | 426 | 13% | 38% | 37% | 4 | 73% | 54% | 55% |
| Postural assessment | 1 | Part 2 | 275 | 30% | 60% | 33% | 4 | 87% | 42% | 89% |
| Postural assessment | 1 | Part 3 | 322 | 21% | 40% | 30% | 4 | 86% | 43% | 56% |
| Evaluation of Muscle Weakness | 2 | Part 1 | 189 | 37% | 46% | 41% | 4 | 92% | 70% | 63% |
| Evaluation of Muscle Weakness | 2 | Part 2 | 460 | 36% | 32% | 13% | 4 | 93% | 25% | 68% |
| Goniometry | 2 | Part 1 | 296 | 72% | 16% | 36% | 4 | 93% | 62% | 73% |
| Goniometry | 2 | Part 2 | 221 | 66% | 25% | 33% | 4 | 89% | 52% | 78% |
| The Shoulder (osseous structures) | 3 | Part 1 | 294 | 65% | 10% | 43% | 4.5 | 94% | 68% | 76% |
| The Shoulder (articular structures) | 3 | Part 2 | 169 | 78% | 12% | 37% | 4 | 96% | 63% | 80% |
| The Shoulder (muscular structures) | 3 | Part 3 | 838 | 71% | 3% | 26% | 4 | 74% | 54% | 61% |
| The Elbow, Wrist & Hand (osseous structures) | 4 | Part 1 | 501 | 85% | 16% | 27% | 4 | 84% | 64% | 88% |
| The Elbow, Wrist & Hand (articular structures) | 4 | Part 2 | 304 | 84% | 21% | 35% | 4 | 92% | 59% | 48% |
| The Elbow, Wrist & Hand (muscular structures) | 4 | Part 3 | 793 | 69% | 19% | 21% | 4 | 64% | 47% | 71% |
| The Hip Complex (osseous structures) | 5 | Part 1 | 236 | 53% | 0% | 33% | 4.5 | 91% | 63% | 80% |
| The Hip Complex (articular structures) | 5 | Part 2 | 57 | 51% | 0% | 27% | 5 | 105% | 60% | 70% |
| The Hip Complex (muscular structures) | 5 | Part 3 | 332 | 22% | 0% | 27% | 5 | 84% | 54% | 71% |
| The Knee Complex (osseous structures) | 6 | Part 1 | 216 | 68% | 3% | 27% | 4 | 85% | 52% | 60% |
| The Knee Complex (articular structures) | 6 | Part 2 | 257 | 68% | 14% | 26% | 4 | 99% | 53% | 78% |
| The Knee Complex (muscular structures) | 6 | Part 3 | 380 | 53% | 4% | 25% | 4 | 78% | 45% | 69% |
| The Ankle Complex (osseous structures) | 7 | Part 1 | 205 | 66% | 4% | 25% | 5 | 100% | 53% | 77% |
| The Ankle Complex (articular structures) | 7 | Part 2 | 134 | 66% | 3% | 21% | 4 | 112% | 48% | 70% |
| The Ankle Complex (muscular structures) | 7 | Part 3 | 241 | 80% | 0% | 22% | 4 | 108% | 47% | 45% |
| The Foot Complex (osseous structures) | 10 | Part 1 | 128 | 47% | 8% | 28% | 4.5 | 98% | 53% | 93% |
| The Foot Complex (articular structures) | 10 | Part 2 | 214 | 87% | 1% | 25% | 4 | 93% | 54% | 68% |
| The Foot Complex (muscular structures) | 10 | Part 3 | 371 | 78% | 1% | 18% | 4 | 78% | 47% | 45% |
| The Brachial Plexus | 11 | Part 1 | 232 | 46% | 5% | 23% | 4 | 95% | 41% | 88% |
| The Lumbosacral Plexus | 11 | Part 2 | 374 | 20% | 5% | 17% | 3.5 | 82% | 36% | 71% |

*(Continued)*

**Table 2.** (Continued)

| | Week | Segment | Duration (s) | Signals | Embodiment | Affective | | Behavioural | Cognitive | |
| | | | | | | Survey submissions | Median survey Score | Audience retention | Quiz attempts | Average quiz scores |
|---|---|---|---|---|---|---|---|---|---|---|
| Human Gait | 12 | Part 1 | 228 | 43% | 20% | 26% | 4.5 | 92% | 49% | 69% |
| Human Gait | 12 | Part 2 | 281 | 46% | 12% | 21% | 4 | 94% | 47% | 53% |
| Human Gait | 12 | Part 3 | 332 | 52% | 13% | 20% | 4 | 95% | 49% | 56% |

Quiz attempts and survey submissions are expressed as a percentage of the overall number of students who had access to the learning units. Audience retention values exceed 100% when learners watch a multimedia lecture (or parts of a multimedia lecture) multiple times.

adj. $R^2$ = .3. Only embodiment made a significantly significant contribution to the prediction, $p$ = .001.

The cumulative odds ordinal logistic model was not statistically significant in predicting the median score on the Likert survey that accompanied each lecture, $\chi^2(3)$ = 5.003, p = .172.

**Behavioural engagement.** The multiple regression model statistically significantly predicted audience retention, $F(3, 31)$ = 15.992, $p$ < .0005, adj. $R^2$ = .56. Only duration made a significantly significant contribution to the prediction, $p$ < .05.

**Cognitive engagement.** The multiple regression model for predicting the number of attempts at each quiz that accompanied each lecture was not considered statistically significant at our a priori alpha $F(3, 31)$ = 3.756, $p$ = .022, adj. $R^2$ = .29. Within this model, signalling (p = 0.005) and embodiment (p = 0.024) each made a significantly significant contribution to the prediction.

The multiple regression model was not statistically significant in predicting the average score on the quizzes that accompanied each lecture, $F(3, 31)$ = 0.327, $p$ = .806, adj. $R^2$ = -.07.

Regression coefficients and standard errors for each of the regression models described above can be found in Tables 3–5.

## Discussion

### Summary of findings

The aim of this study was to evaluate the effect of multimedia lecture style on student engagement. For the purposes of this study, 'style' was defined according to the extent to which selected principles from Mayer's theory of multimedia instruction were implemented across a catalogue of multimedia lectures. According to this theory, there are three kinds of demands on cognitive capacity during learning [14]. These demands correspond with three kinds of instructional design targets: to reduce extraneous cognitive processing, to manage essential

**Table 3. Multiple regression results for the number of survey submissions.**

| Survey submissions | B | 95% CI for B | | SE B | ß | $R^2$ | Adjusted $R^2$ | P |
|---|---|---|---|---|---|---|---|---|
| | | LL | UL | | | 0.37 | 0.3 | .005 |
| (Constant) | 0.22 | 0.11 | 0.32 | 0.05 | | | | |
| Segmenting | 0 | 0 | 0 | 0 | -0.254 | | | |
| Signals | 0.1 | -0.03 | 0.24 | 0.07 | 0.258 | | | |
| Embodiment | 0.33 | 0.15 | 0.5 | 0.09 | 0.643 | | | |

Note. Model = "Enter" method in SPSS Statistics. B = unstandardised regression coefficient; CI = confidence interval; LL = lower limit; UL = upper limit; SE B = standard error of the coefficient; ß = standardized coefficient; $R^2$ = coefficient of determination

**Table 4. Multiple regression results for audience retention.**

| Audience Retention | B | 95% CI for B | | SE B | ß | $R^2$ | Adjusted $R^2$ | P |
|---|---|---|---|---|---|---|---|---|
| | | LL | UL | | | 0.63 | 0.59 | < .0005 |
| (Constant) | 1.011 | 0.914 | 1.107 | 0.047 | | | | |
| Segmenting | 0 | -0.001 | 0 | 0 | -0.757 | | | |
| Signals | 0.081 | -0.046 | 0.209 | 0.062 | 0.167 | | | |
| Embodiment | -0.058 | -0.22 | 0.103 | 0.079 | -0.095 | | | |

Note. Model = "Enter" method in SPSS Statistics. *B* = unstandardised regression coefficient; CI = confidence interval; LL = lower limit; UL = upper limit; *SE B* = standard error of the coefficient; ß = standardized coefficient; $R^2$ = coefficient of determination.

cognitive processing, and to foster generative cognitive processing. The theory proposes 15 principles which can be manipulated to achieve these targets [13]; the current study used transaction log analyses to evaluate the effect of the segmenting, signalling and embodiment principles on student engagement, whilst holding the remaining 12 principles constant between multimedia lectures. Engagement behaviours were evaluated in three domains [43]: affective engagement was determined by students' responses on a Likert-style survey that accompanied each multimedia lecture; behavioural engagement was evaluated through Web log data provided YouTube analytics [42]; cognitive engagement was determined by students' submissions on a quiz that accompanied each multimedia lecture. Separate regression models for each domain of engagement revealed that embodiment, segmenting and signalling increased affective, behavioural and cognitive engagement with multimedia lectures, respectively, confirming our first and second experimental hypotheses. However, the regression model for MCQ scores, which were used as a surrogate for knowledge retention, was non-significant; this contradicted our third experimental hypothesis.

## Empirical contributions

This study is unique in several respects. First, to the author's knowledge, this is the first study to differentially 'stack' selected principles from the CTML across a catalogue of multimedia instructional videos, and to longitudinally evaluate the effect of this on student engagement using a transaction log analysis methodology. Namely, the signalling principle was used to reduce extraneous cognitive processing [19], the segmenting principle was used to manage essential cognitive processing [21] and the embodiment principle was used to foster generative cognitive processing [22]. These techniques were applied to various extents across a catalogue of 32 multimedia lectures, and in this way, it was possible to evaluate the association between overall 'style' (or the extent to which these techniques were applied for a given piece of

**Table 5. Multiple regression results for the number of attempts made the quizzes that accompanied each multimedia lecture.**

| Quiz attempts | B | 95% CI for B | | SE B | ß | $R^2$ | Adjusted $R^2$ | P |
|---|---|---|---|---|---|---|---|---|
| | | LL | UL | | | 0.29 | 0.21 | .02 |
| (Constant) | 0.369 | 0.228 | 0.51 | 0.069 | | | | |
| Segmenting | 0 | 0 | 0 | 0 | -0.192 | | | |
| Signals | 0.277 | 0.089 | 0.464 | 0.092 | 0.538 | | | |
| Embodiment | 0.277 | 0.039 | 0.514 | 0.116 | 0.427 | | | |

Note. Model = "Enter" method in SPSS Statistics. *B* = unstandardised regression coefficient; CI = confidence interval; LL = lower limit; UL = upper limit; *SE B* = standard error of the coefficient; ß = standardized coefficient; $R^2$ = coefficient of determination.

multimedia instruction) and students' engagement behaviours. Studies in the field of multimedia instruction typically assign learners to a control group or an experimental group that is identical except that the experimental group of learners receives a lesson that contains the to-be-tested principle while a control group of learners receives an otherwise identical lesson that lacks the to-be-tested principle [23]. This creates a false dichotomy in which individual principles are deemed to be effective or ineffective, ignoring any underlying association between engagement behaviours and one or more of the design principles.

The second unique characteristic of this study is that it follows the principle of representative design [44] in that the instructional material was directly relevant to students' desired profession. Specifically, the multimedia lectures were delivered to students enrolled in a University module related to 'Functional Anatomy & Kinesiology'. Learning material for this module related to the knowledge and function of "human osseous, joint, muscular and neural structures". The learning outcomes for this module include a basic competency in upper and lower limb muscle strength, flexibility and range-of-motion assessment, the analysis of human posture and gait, and an ability to accurately palpate body structures. The module is a core part of students learning for the subdisciplines of musculoskeletal physical therapy and applied exercise science, and as such, it is likely that students had a high degree of motivation to learn during multimedia instruction. The learning units in the aforementioned research in the field of multimedia instruction typically only exist as experimental entities with no real-world relevance to either the control or intervention groups.

The third unique aspect of this study is in quantifying the relationship between the style of a piece of multimedia instruction (or the extent to which the above-listed techniques were implemented) on engagement in three domains congruently [43]. In the science of instruction, learning processes–including affective, behavioural and cognitive–are often assessed by subjective measures such as questionnaires, which are administered after or during a multimedia lesson [45]. The validity of subjective measures has been questioned however, as the administration of a questionnaire during a lesson may impact cognitive processes while recall bias may impact results if the questionnaire is administered after the lesson [46, 47]. In contrast, objective measures such as eye-tracking [48], Web-log data [31, 35, 49], dual-task paradigms [50], brain activity [51], and biometrics like heart rate variability [52] are increasingly popular in extant literature and offer the advantage of directly tapping into the learner's behaviour during learning. By combining outcomes, this longitudinal analysis adds to the robustness of the research concerning the impact of the segmenting, signalling and embodiment principles on the cognitive processes of learning.

The final novelty of this research is that the effect of these principles on measures of engagement were evaluated longitudinally in a single cohort of 92 students from one of two degree programmes (Physiotherapy or Health & Performance Science) all of whom were enrolled in one module for which the material was all developed and delivered by one instructor. Therefore, the potentially confounding effects of having different instructors deliver heterogenous learning material in varying ways to distinct cohorts of students (each with diverse affective, motivational, metacognitive or social learning processes) were mitigated.

## Practical implications

These results should be a welcome addition to efforts for developing empirically derived principles for instructional design to better help people learn. Based on the multiple linear regression analyses, affective engagement (*A*, as determined by the number of submissions,

expressed as a % of group size) with a multimedia lecture can be calculated with the following formula:

$$A = 0.216 + (0.103 \times [\% \text{ of video where signals are present}]) + (0.325 \times [\% \text{ of video where there is a talking head on screen}])$$

Behavioural engagement (B, as determined by audience retention, or the average view duration of a multimedia lecture, expressed as a percentage of its overall length) with a multimedia lecture can be calculated with the following formula:

$$B = 1.011 + (0.081 \times [\% \text{ of video where signals are present}]) - (0.058 \times [\% \text{ of video where there is a talking head on screen}])$$

Cognitive engagement (C, as determined by the number of MCQ attempts, expressed as a % of group size) with a multimedia lecture can be calculated with the following formula:

$$C = 0.369 + (0.277 \times [\% \text{ of video where signals are present}]) + (0.277 \times [\% \text{ of video where there is a talking head on screen}])$$

Rather than using these formulae to define the parameters for a multimedia lecture, they should be used as a guide for instructors, and demonstrate the value of incorporating the segmenting, signalling and embodiment principles within their multimedia lectures.

## Theoretical implications

While our results confirmed our first and secondary experimental hypotheses, that the regression model for test scores was non-significant was surprising and contradicted our third hypothesis. This finding also contradicts the results of several meta-analyses which have demonstrated the effectiveness of signalling [53], segmenting [54] and embodiment [55] for test performance. There are a number of possible explanations for this. First, all of the multimedia lectures included in the current study were created to facilitate student learning, albeit to varying extents. Each lecture employed many of the other twelve principles of multimedia learning (including the multimedia, modality, personalisation, voice, image, coherence, spatial contiguity, temporal contiguity and redundancy principles), and none of the lectures were designed to overtly challenge or inhibit students' learning. The dataset did not include any lecture which did not stack at least two of the signalling, segmenting or embodiment principles, either. Together, the combination of principles within each multimedia lecture may have underpowered our analysis for observing potentially negative effects on students' test scores. While we acknowledge this as a limitation of the study, it should be noted that including such lectures may have unfairly disadvantaged students and could be considered unethical in a university cohort.

Another possible explanation for the lack of an effect on test scores is that, because the content of all the multimedia lectures was relevant to students' course material, they may have had a higher baseline level of generative cognitive processing to 'overcome' the challenges posed by any increases in extraneous cognitive processing (where multimedia lectures were longer), where essential cognitive processing was not managed (by omitting signals) or where generative processing was not encouraged through overt displays of embodiment. Generative processing, analogous to germane cognitive load in cognitive load theory [56, 57], is caused by the learner's motivation to learn, and may have already been primed in students prior to exposure to each of the multimedia lectures in the catalogue, limiting the sensitivity of the analyses for identifying negative test effects. This raises the possibility that while there was no direct effect of the signalling, segmenting or embodiment principles on test performance, because these

principles did increase affective and behavioural engagement that the effect on cognitive engagement might be further down the 'pedagogical chain'. Further research is warranted to establish whether affective and behavioural engagement might predict cognitive engagement.

Finally, several "boundary conditions" have been identified for each of the signalling, segmenting and embodiment principles. For all three principles, a prominent boundary condition that has been identified which has previously been shown to undermine the effectiveness of each of the segmenting [58], signalling [59] and embodiment [60] principles is in instances where learners had some prior knowledge of the learning material; a pre-requisite for enrolment in the module for which the multimedia lectures in the current study were prepared was completion of a module in basic human anatomy. That a large proportion of the learning material covered in the multimedia lectures for 'Functional Anatomy & Kinesiology' concerned basic human anatomy (including anatomical terminology, anatomical site identification and the structural organisation of the human body), may have compromised the effect of these principles on test scores in this cohort.

## Limitations and future directions

Despite its strengths, this study is not without limitations. As noted previously, because we evaluated engagement behaviours in students enrolled in a university module, while this enabled us to control for potentially confounding effects (like sample heterogeneity, students' baseline knowledge, their motivation to learn and the relevance of the learning material to them), this may have also undermined the external validity of our results. Future research should clarify whether our findings can be replicated among separate cohorts of learners (perhaps further into their degree programs), with different instructors teaching alternative course syllabi.

Next, while transaction log analysis is a shrewd way to elicit actual user behavioural data [31], those data can only be analysed at an aggregate level [35]. Therefore, our findings provide limited insights into individual students' engagement behaviours whilst watching lectures from the catalogue.

Based on the assumption that transaction log analysis offers novel insights into actual user behaviour during learning, future research should also replicate this research design to evaluate the other principles from the CTML. In particular, we believe that transaction log analysis would be suited to evaluating the effectiveness of the "immersion principle" which presently states that "people do not necessarily learn better in 3D immersive virtual reality than with a corresponding 2D desktop presentation" [61]; specifically, transaction log analysis might provide novel insights into how users' behaviours are altered in virtual or augmented reality compared with non-immersive learning environments. Indeed such an analysis could be undertaken in a similar fashion to this investigation using the analytics provided by YouTube studio [42].

Furthermore, we also believe that transaction log analysis could provide the empirical foundation for the development of new principles. For instance, Mayer and colleagues discuss the lack of effectiveness of "seductive" techniques like adding music or sound effects to multimedia lectures [62], however this is a widely adopted practice among popular content creators (e.g. Crash Course [2022], which has amassed >1bn views to date [63]). Future studies could reconcile this apparent contradiction.

Finally, due to the design of the study, we were unable to infer any causal relationships between the principles of instructional design and the measures of engagement.

## Conclusions

In conclusion, this study revealed the effectiveness of the signalling, segmenting and embodiment principles for increasing students' affective and behavioural engagement with

multimedia lectures. Instructors should be encouraged to break multimedia lectures into shorter 'chunks' to help learners manage essential cognitive processing, to incorporate visual and verbal cues (i.e., signals) to reduce extraneous cognitive processing to direct learners towards important learning material, and to incorporate a video of themselves enthusiastically narrating the learning material within a multimedia lecture to foster generative cognitive processing in the learner. Further research is required to determine whether this study and these results can be replicated for different topics and cohorts of students, to investigate the relative effectiveness of the other 12 principles proposed in the CTML, and to further evaluate the relationship between affective, behavioural and cognitive engagement during multimedia learning.

## Author Contributions

**Data curation:** Cailbhe Doherty.

**Formal analysis:** Cailbhe Doherty.

**Investigation:** Cailbhe Doherty.

**Methodology:** Cailbhe Doherty.

**Project administration:** Cailbhe Doherty.

**Resources:** Cailbhe Doherty.

**Software:** Cailbhe Doherty.

**Validation:** Cailbhe Doherty.

**Visualization:** Cailbhe Doherty.

**Writing – original draft:** Cailbhe Doherty.

**Writing – review & editing:** Cailbhe Doherty.

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
