## [Decision Letter · Decision Letter 0]

27 Jun 2022

PONE-D-22-10658A web log analysis investigating the relationship between the instructional design features of a catalogue of multimedia lectures and learners’ engagement behaviours.PLOS ONE

Dear Dr. Doherty,

Thank you for submitting your manuscript to PLOS ONE. After careful consideration, we feel that it has merit but does not fully meet PLOS ONE’s publication criteria as it currently stands. Therefore, we invite you to submit a revised version of the manuscript that addresses the points raised during the review process.

We look forward to receiving your revised manuscript.

Kind regards,

Sun Kyung Kim

Academic Editor

PLOS ONE

Journal Requirements:

Reviewers' comments:

Reviewer's Responses to Questions

**Comments to the Author**

1. Is the manuscript technically sound, and do the data support the conclusions?

Reviewer #1: Yes

Reviewer #2: Yes

Reviewer #3: Yes

2. Has the statistical analysis been performed appropriately and rigorously? 

Reviewer #1: Yes

Reviewer #2: Yes

Reviewer #3: Yes

3. Have the authors made all data underlying the findings in their manuscript fully available?

Reviewer #1: Yes

Reviewer #2: Yes

Reviewer #3: Yes

4. Is the manuscript presented in an intelligible fashion and written in standard English?

Reviewer #1: No

Reviewer #2: Yes

Reviewer #3: Yes

5. Review Comments to the Author

Reviewer #1: Authors have well defined problem area to identify effectiveness of signaling, segmentation and embodiment while delivering lectures in the form of multimedia. Study is well experimented with suitable and sufficient number of participants and proved hypothesis which are mentioned in the papers. Authors have considered lot of statistical parameters and performed relevant tests to analyze the captured data using logs. However the major concern in the paper, is the formatting of the paper. Also there is a need to defining sections and subsections properly in the paper. Paper is written technically well however English corrections need to be performed.

Reviewer #2: In this paper, students’ engagement behaviors were evaluated using the transaction log analysis with instructional design principles from CTML. And the log data were statistically analyzed using multiple regression methods. The author contributed to related research by providing a more quantitative method of evaluation than previous study.

However, I think that this paper needs minor revision to some of the following issues.

1. Remove unnecessary sentences from the abstract

- The abstract is too verbose. Only the core content should be summarized.

Ex) In 30-31 lines, It seems unnecessary for me to describe the statement “This lecture enhanced my learning/understanding of course content” and the scale score in the abstract.

2. Awkward sentences

- In 105-108 lines, I do not understand the relationship between "the principles of the CMTL being evaluated in isolation" and "learners are randomly assigned". The author needs a supplementary explanation for this part.

- In 316-330 lines, the sentence “that accompanied each multimedia lecture” repeats over and over again. Even if this sentence is deleted, there seems to be no difficulty for the reader to understand this chapter.

3. Lack of explanation

- In table 1, can the amount of time signals and the amount of time the lecturer be collected automatically? I don't think it can be collected automatically. If automatic collection is not possible, it cannot be said to be the transaction log. The author should explain further about this.

- In table 1, what is the meaning of the number of attempts in the quiz? Is it the number of retries for a student to get 100 points for the quiz?

- The note in table 1 looks good to be moved to table 2. I checked the value over 100% on Table 2.

- In 336 line, What do you mean by the parameter values in F(3, 31)? I couldn't find the meaning of this value in the paper.

4. others

- I can't really see the content in the figure 1. It should be replaced with a high-resolution image.

- The page number of the reference is not consistent. It is "401-414" correct, not "401-14" in 381 line. The author needs a lot of corrections other than that.

Reviewer #3: Thanks for inviting me to review this paper, which is interesting and solid. I recommend minor revision, and I have the following suggestions:

The title is a bit too long, and hard to understand. I suggest the author to consider summary the study in a clear and simple way, and write a more direct title without so many details in it.

The method, Transaction log analysis. I don’t fully understand why this method is better than other methods, such as process mining, network analysis, etc. I suggest the author to provide a systematic review of relevant methods then introduce this Transaction log analysis method.

There are also some other very relevant papers are not included here as references, such as:

Fan, Y., Matcha, W., Uzir, N. A. A., Wang, Q., & Gašević, D. (2021). Learning analytics to reveal links between learning design and self-regulated learning. International Journal of Artificial Intelligence in Education, 31(4), 980-1021.

Pinder, R. A., Davids, K., Renshaw, I., & Araújo, D. (2011). Representative learning design and functionality of research and practice in sport. Journal of Sport and Exercise Psychology, 33(1), 146-155.

The discussion is well written, but also very long, I suggest the author to further organise it a bit, such as adding more meaningful subheading to highlight the points you would like to address.

6. PLOS authors have the option to publish the peer review history of their article (what does this mean?). If published, this will include your full peer review and any attached files.

Reviewer #1: No

Reviewer #2: No

Reviewer #3: No

---

## [Author Response · Author response to Decision Letter 0]

28 Jun 2022

Response to reviewers

Reviewer #1: Authors have well defined problem area to identify effectiveness of signaling, segmentation and embodiment while delivering lectures in the form of multimedia. Study is well experimented with suitable and sufficient number of participants and proved hypothesis which are mentioned in the papers. Authors have considered lot of statistical parameters and performed relevant tests to analyze the captured data using logs. However the major concern in the paper, is the formatting of the paper. Also there is a need to defining sections and subsections properly in the paper. Paper is written technically well however English corrections need to be performed.

AUTHOR RESPONSE: Many thanks for taking the time to review this manscript and for your positive appraisal. In response to your comments and those of the other reviewers, the paper has been rewritten to improve clarity and organisation. Spelling and grammar have been revised throughout. Please see the resubmission document for details

Reviewer #2: In this paper, students’ engagement behaviors were evaluated using the transaction log analysis with instructional design principles from CTML. And the log data were statistically analyzed using multiple regression methods. The author contributed to related research by providing a more quantitative method of evaluation than previous study.

AUTHOR RESPONSE: Thank you for the time you have taken to review the paper. The manuscript has been revised in line with your comments, details for which are presented below.

However, I think that this paper needs minor revision to some of the following issues.

1. Remove unnecessary sentences from the abstract

- The abstract is too verbose. Only the core content should be summarized.

Ex) In 30-31 lines, It seems unnecessary for me to describe the statement “This lecture enhanced my learning/understanding of course content” and the scale score in the abstract.

AUTHOR RESPONSE: The abstract has been rewritten to improve clarity:

“The purpose of this transaction log analysis was to evaluate university students’ engagement behaviours with a catalogue of multimedia lectures. These lectures incorporated selected instructional design principles from the cognitive theory of multimedia learning (CTML). Specifically, thirty-two multimedia lectures which differentially employed the signalling, segmenting and embodiment principles from the CTML were delivered to a cohort of 92 students throughout an academic trimester. Engagement with each multimedia lecture was measured in three domains: affective engagement was measured using a Likert-style survey that accompanied each multimedia lecture; behavioural engagement was measured using the web logs provided by YouTube Studio analytics (average watch time); cognitive engagement was measured using students’ average score on a quiz that accompanied each multimedia lecture. Separate multiple linear regression analyses for measures of affective, behavioural and cognitive engagement revealed that multimedia lectures that ‘stacked’ the instructional design principles of high embodiment (whereby the lecture was interspersed with clips of an enthusiastic onscreen instructor), segmenting (where lectures were divided into shorter, user-paced segments) and signalling (where onscreen labels highlighted important material) increased measures of engagement, including overall watch time, number of survey submission and number of quiz attempts (P < 0.05). There was no association between any of the tested principles and students’ quiz scores or their responses on the Likert-style survey. This study adds to the available literature demonstrating the effectiveness of the signalling, segmenting and embodiment principles for increasing learner engagement with multimedia lectures.”

2. Awkward sentences

- In 105-108 lines, I do not understand the relationship between "the principles of the CMTL being evaluated in isolation" and "learners are randomly assigned". The author needs a supplementary explanation for this part.

AUTHOR RESPONSE: This section has been rewritten to improve clarity:

“First, principles from the CTML tend to be evaluated individually; this does not reflect “real world” scenarios where learning materials will often incorporate several principles (like the signalling, segmenting and embodiment principles) together in a single piece of multimedia (1). Extant research also rarely evaluates this kind of content in real-world contexts, where learning material is part of a formal degree program (2).” 

- In 316-330 lines, the sentence “that accompanied each multimedia lecture” repeats over and over again. Even if this sentence is deleted, there seems to be no difficulty for the reader to understand this chapter.

AUTHOR RESPONSE: Amended as requested.

3. Lack of explanation

- In table 1, can the amount of time signals and the amount of time the lecturer be collected automatically? I don't think it can be collected automatically. If automatic collection is not possible, it cannot be said to be the transaction log. The author should explain further about this.

AUTHOR RESPONSE: The ‘multimedia characteristics’ were the independent variables in our analysis; the transaction logs only related to the dependent variables (i.e., quiz attempts, quiz scores, average watch time, survey attempts, survey scores). All of the dependent variables were collected automatically; this is typical of transaction log analysis.

- In table 1, what is the meaning of the number of attempts in the quiz? Is it the number of retries for a student to get 100 points for the quiz?

AUTHOR RESPONSE: This has been clarified as follows:

“The number of attempts completed by learners on the quiz associated with a multimedia lecture. Expressed as a percentage of the number of learners who had access to the quiz (i.e. the total number of learners who had access to the learning units [N = 92]). Each student could only attempt each quiz once.”

- The note in table 1 looks good to be moved to table 2. I checked the value over 100% on Table 2.

AUTHOR RESPONSE: Amended as requested.

- In 336 line, What do you mean by the parameter values in F(3, 31)? I couldn't find the meaning of this value in the paper.

AUTHOR RESPONSE: This is standard output for multiple regression. ‘F’ indicates that a comparison was made to an F-distribution (F-Test); ‘3’ indicates the degrees of freedom in the model; ‘31’ indicates the residual degrees of freedom (aka error) while ‘15.992’ indicates the obtained value of the F-statistic.

A nice summary of these annotations (and how to conduct multiple regression) is presented here:

https://statistics.laerd.com/premium/spss/mr/multiple-regression-in-spss-16.php

4. others

- I can't really see the content in the figure 1. It should be replaced with a high-resolution image.

AUTHOR RESPONSE: Resolution of Figure 1 has been improved as requested.

- The page number of the reference is not consistent. It is "401-414" correct, not "401-14" in 381 line. The author needs a lot of corrections other than that.

AUTHOR RESPONSE: The references have been checked and corrected as requested.

Reviewer #3: Thanks for inviting me to review this paper, which is interesting and solid. I recommend minor revision, and I have the following suggestions:

AUTHOR RESPONSE: Many thanks for your input and for the time you have taken to review this paper. I have made a series of amendments in response to the comments you raise, details of which are presented below.

The title is a bit too long, and hard to understand. I suggest the author to consider summary the study in a clear and simple way, and write a more direct title without so many details in it.

AUTHOR RESPONSE: The title has been amended as follows:

“An investigation into the relationship between multimedia lecture design and learners’ engagement behaviours using web log analysis.”

The method, Transaction log analysis. I don’t fully understand why this method is better than other methods, such as process mining, network analysis, etc. I suggest the author to provide a systematic review of relevant methods then introduce this Transaction log analysis method.

AUTHOR RESPONSE: While a full systematic review is beyond the scope of this paper, we have added additional material outlining what transaction log analysis is and relevant prior research to further explain this methodological approach. 

“Advancing on traditional measures likes interviews and surveys, transaction log analysis is a broad methodological approach used to evaluate the electronic records of interactions that have occurred between a system and users of that system (3). Alternately referred to as web-log analysis, search log analysis and digital trace data (4), transaction log analysis has been in use since at least 1967 (5) and has been published in peer reviewed research since 1975 (6), this methodology has undergone a resurgence in recent times having been made more accessible by the widespread availability of user-facing analytics APIs by video sharing and social media platforms like YouTube and Facebook. 

For transaction log analysis, behaviour is the essential construct and is defined as any observable activity of a person, team, organization, or system; session duration or number of clicks are typical variables that researchers are interested in (7). From the perspective of measuring learners’ behaviour, the advantage of transaction log analysis over interviews, surveys or think-aloud exercises is that it is an unobtrusive method of data collection as it does not involve directly interfacing with learners (3).

Transaction logs are widely captured in university settings: e-resource usage, computer logins, attendance and activity in virtual learning environments have been evaluated to discern how students learn.(8) Despite this, universities have been slow to fully utilise the potential of transaction log analysis to shine a light on information-interaction behaviours on content delivery platforms such as YouTube. Despite all the benefits of the Web for eLearning,(9, 10) social media-based education,(11) virtual reality classrooms(12) and as a channel for academic institutions to supply their students with content in a socially distant society,(13) there is a dearth of research that leverages transaction log analysis methods to evaluate user behaviour during multimedia instruction, particularly when evaluated under the paradigm of the CTML.“

There are also some other very relevant papers are not included here as references, such as:

Fan, Y., Matcha, W., Uzir, N. A. A., Wang, Q., & Gašević, D. (2021). Learning analytics to reveal links between learning design and self-regulated learning. International Journal of Artificial Intelligence in Education, 31(4), 980-1021.

Pinder, R. A., Davids, K., Renshaw, I., & Araújo, D. (2011). Representative learning design and functionality of research and practice in sport. Journal of Sport and Exercise Psychology, 33(1), 146-155.

AUTHOR RESPONSE: These references have been incorporated into our introduction and discussion. However we would offer that, while related in that it discussed learning tactics and strategies, the first paper relied on a largely ML-focused analysis of MOOC data. Our dataset was not large enough to train an ML system.

The second paper on representative design is certainly relevant and is a strength of the current study-we believe our study is unique in that it evaluate learners for whom the learning material was part of their degree program, rather than being an empirical construct.

The discussion is well written, but also very long, I suggest the author to further organise it a bit, such as adding more meaningful subheading to highlight the points you would like to address.

AUTHOR RESPONSE: The discussion has been reorganised and new headings have been added as requested. Please see the document for details.

1. Mayer R. Multimedia Learning (Chapter 3). 3rd ed: Cambridge University Press; 2020.

2. Guo P, Kim J, Rubin R. How video production affects student engagement: an empirical study of MOOC videos. Proceedings of the first ACM conference on Learning @ scale conference; Atlanta, Georgia, USA: Association for Computing Machinery; 2014. p. 41–50.

3. Spink A, Jansen B, Taksa I. Handbook of Web Log Analysis: Information Science Reference - Imprint of: IGI Publishing; 2008.

4. Fan Y, Matcha W, Uzir NaA, Wang Q, Gašević D. Learning Analytics to Reveal Links Between Learning Design and Self-Regulated Learning. International Journal of Artificial Intelligence in Education. 2021;31(4):980-1021.

5. Meister D, Sullivan DJ. Evaluation of user reactions to a prototype on-line information retrieval system. NASA CR-918. NASA Contract Rep NASA CR. 1967:1-58.

6. Penniman W. A stochastic process analysis of online user behavior. Annual Meeting of the American Society for Information Science; 26-30 October; Washington, DC.1975.

7. Doherty C, Joorabchi A, Megyesi P, Flynn A, Caulfield B. Physiotherapists' Use of Web-Based Information Resources to Fulfill Their Information Needs During a Theoretical Examination: Randomized Crossover Trial. Journal of medical Internet research. 2020;22(12):e19747.

8. Showers B. Library Analytics and Metrics: Using Data to Drive Decisions and Services. UK: Facet Publishing; 2015.

9. Ruiz JG, Mintzer MJ, Leipzig RM. The impact of E-learning in medical education. Academic medicine : journal of the Association of American Medical Colleges. 2006;81(3):207-12.

10. Harder B. Are MOOCs the future of medical education? BMJ (Clinical research ed). 2013;346:f2666.

11. Cheston CC, Flickinger TE, Chisolm MS. Social media use in medical education: a systematic review. Academic medicine : journal of the Association of American Medical Colleges. 2013;88(6):893-901.

12. Levac D, Glegg SM, Sveistrup H, Colquhoun H, Miller PA, Finestone H, et al. A knowledge translation intervention to enhance clinical application of a virtual reality system in stroke rehabilitation. BMC health services research. 2016;16(1):557.

13. Sandhu P, de Wolf M. The impact of COVID-19 on the undergraduate medical curriculum. Med Educ Online. 2020;25(1):1764740.

---

## [Decision Letter · Decision Letter 1]

1 Aug 2022

An investigation into the relationship between multimedia lecture design and learners’ engagement behaviours using web log analysis.

PONE-D-22-10658R1

Dear Dr. Doherty 

We’re pleased to inform you that your manuscript has been judged scientifically suitable for publication and will be formally accepted for publication once it meets all outstanding technical requirements.

Kind regards,

Sun Kyung Kim

Academic Editor

PLOS ONE

Additional Editor Comments (optional):

Reviewers' comments:

Reviewer's Responses to Questions

**Comments to the Author**

1. If the authors have adequately addressed your comments raised in a previous round of review and you feel that this manuscript is now acceptable for publication, you may indicate that here to bypass the “Comments to the Author” section, enter your conflict of interest statement in the “Confidential to Editor” section, and submit your "Accept" recommendation.

Reviewer #2: All comments have been addressed

2. Is the manuscript technically sound, and do the data support the conclusions?

Reviewer #2: Yes

3. Has the statistical analysis been performed appropriately and rigorously? 

Reviewer #2: (No Response)

4. Have the authors made all data underlying the findings in their manuscript fully available?

Reviewer #2: Yes

5. Is the manuscript presented in an intelligible fashion and written in standard English?

Reviewer #2: Yes

6. Review Comments to the Author

Reviewer #2: (No Response)

7. PLOS authors have the option to publish the peer review history of their article (what does this mean?). If published, this will include your full peer review and any attached files.

Reviewer #2: No

---

## [Editor Report · Acceptance letter]

4 Aug 2022

PONE-D-22-10658R1 

An investigation into the relationship between multimedia lecture design and learners’ engagement behaviours using web log analysis. 

Dear Dr. Doherty:

I'm pleased to inform you that your manuscript has been deemed suitable for publication in PLOS ONE. Congratulations! Your manuscript is now with our production department. 

Kind regards, 

on behalf of

Professor Sun Kyung Kim 

Academic Editor

PLOS ONE